Natural variation in teosinte at the domestication locus teosinte branched1 (tb1)

Vann Laura 1
Kono Thomas 1 2
Pyhäjärvi Tanja 1 3
Hufford Matthew B. 1 4 mhufford@iastate.edu
Ross-Ibarra Jeffrey 1 5 rossibarra@ucdavis.edu
1 Department of Plant Sciences, University of California , Davis, CA , USA
2 Department of Agronomy and Plant Genetics, University of Minnesota , Twin Cities, Minneapolis, MN , USA
3 Department of Biology, University of Oulu , Oulu , Finland
4 Department of Ecology, Evolution, and Organismal Biology, Iowa State University , Ames, Iowa , USA
5 Center for Population Biology and Genome Center, University of California , Davis, CA , USA
Vision Todd
Electronic publication date: 2015 Apr 16
Publication date: 2015
Volume: 3
Electronic Location ID: e900
Received 2014 Dec 10; Accepted 2015 Mar 30
Copyright: © 2015 Vann et al.
Copyright year: 2015
Copyright holder: Vann et al.
License: This is an open access article distributed under the terms of the Creative Commons Attribution License, which permits unrestricted use, distribution, reproduction and adaptation in any medium and for any purpose provided that it is properly attributed. For attribution, the original author(s), title, publication source (PeerJ) and either DOI or URL of the article must be cited.
License URL: https://creativecommons.org/licenses/by/4.0/

Keywords: Transposable element, Domestication, Teosinte, Teosinte branched1, Maize

Funding: Department of Plant Sciences UC Mexus Funding was provided by the Department of Plant Sciences at UC Davis for graduate student research funding to LEV and for research funds supporting the project, and by UC Mexus for a postdoctoral scholar grant to MBH and JR-I. The funders had no role in study design, data collection and analysis, decision to publish, or preparation of the manuscript.

==============================
The teosinte branched1(tb1) gene is a major QTL controlling branching differences between maize and its wild progenitor, teosinte. The insertion of a transposable element (Hopscotch) upstream of tb1 is known to enhance the gene’s expression, causing reduced tillering in maize. Observations of the maize tb1 allele in teosinte and estimates of an insertion age of the Hopscotch that predates domestication led us to investigate its prevalence and potential role in teosinte. We assessed the prevalence of the Hopscotch element across an Americas-wide sample of 837 maize and teosinte individuals using a co-dominant PCR assay. Additionally, we calculated population genetic summaries using sequence data from a subset of individuals from four teosinte populations and collected phenotypic data using seed from a single teosinte population where Hopscotch was found segregating at high frequency. Genotyping results indicate the Hopscotch element is found in a number of teosinte populations and linkage disequilibrium near tb1 does not support recent introgression from maize. Population genetic signatures are consistent with selection on the tb1 locus, revealing a potential ecological role, but a greenhouse experiment does not detect a strong association between the Hopscotch and tillering in teosinte. Our findings suggest the role of Hopscotch differs between maize and teosinte. Future work should assess tb1 expression levels in teosinte with and without the Hopscotch and more comprehensively phenotype teosinte to assess the ecological significance of the Hopscotch insertion and, more broadly, the tb1 locus in teosinte.

Introduction

Domesticated crops and their wild progenitors provide an excellent system in which to study adaptation and genomic changes associated with human-mediated selection (Ross-Ibarra, Morrell & Gaut, 2007). Plant domestication usually involves a suite of phenotypic changes such as loss of seed shattering and increased fruit or grain size, which are commonly referred to as the ‘domestication syndrome’ (Olsen & Wendel, 2013), and much of the study of domestication has focused on understanding the genetic variation underlying these traits (Olsen & Gross, 2010). Because most domesticates show reduced genetic diversity relative to their wild counterparts, effort has been made to identify agronomically useful variation in crop wild relatives (Flint-Garcia, Bodnar & Scott, 2009). In some instances, the alleles conferring these beneficial traits are bred into domesticates for crop improvement. For example, Oryza rufipogon, the wild progenitor of domesticated rice, has proven useful for the integration of a number of beneficial QTL controlling traits such as grain size and yield into domesticated rice (Kovach & McCouch, 2008). In addition to researching the role of wild alleles in domesticates, researchers have also investigated the role of variation in domesticated taxa in the evolution of feral and weedy populations (Ellstrand et al., 2010). But even though domesticated alleles are often found segregating in wild relatives (Gallavotti et al., 2004; Sigmon & Vollbrecht, 2010), little is known about the ecological role of this variation in natural populations. In this paper we present an ecological genetic analysis of the domestication locus tb1—specifically the domesticated haplotype at tb1—in natural populations of the wild ancestor of domesticated maize.

Maize (Zea mays ssp. mays) was domesticated from the teosinte Zea mays ssp. parviglumis (hereafter, parviglumis) roughly 9,000 B.P. in southwest Mexico (Piperno et al., 2009; Matsuoka et al., 2002). Maize and the teosintes are an attractive system in which to study domestication due to the abundance of genetic tools developed for maize and well-characterized domestication loci (Hufford et al., 2012a; Doebley, 2004; Hufford et al., 2012b). Additionally, large, naturally-occurring populations of both parviglumis and Zea mays ssp. mexicana (hereafter, mexicana) can be found throughout Mexico (Wilkes, 1977; Hufford et al., 2013), with parviglumis distributed in the lowlands of Mexico and mexicana in the highlands. Furthermore, both parviglumis and mexicana occur at high densities and genetic diversity of these taxa is estimated to be high (Hufford et al., 2012a; Ross-Ibarra, Tenaillon & Gaut, 2009).

Many morphological changes are associated with maize domestication, and understanding the genetic basis of these changes has been a focus of maize research for a number of years (Doebley, 2004). One of the most dramatic changes is found in plant architecture: domesticated maize is characterized by a central stalk with few tillers and lateral branches terminating in a female inflorescence, while teosinte is highly tillered and bears tassels (male inflorescences) at the end of its lateral branches. The teosinte branched1 (tb1) gene, a repressor of organ growth, was identified as a major QTL involved in branching (Doebley, Stec & Gustus, 1995) and tillering (Doebley & Stec, 1991) differences between maize and teosinte. A 4.9 kb retrotransposon (Hopscotch) insertion into the upstream control region of tb1 in maize acts to enhance expression of tb1, thus repressing lateral organ growth (Doebley, Stec & Hubbard, 1997; Studer et al., 2011). Dating of the Hopscotch retrotransposon suggests that its insertion predates the domestication of maize, leading to the hypothesis that it was segregating as standing variation in populations of teosinte and increased to high frequency in maize due to selection during domestication (Studer et al., 2011). The effects of the Hopscotch insertion have been studied in maize (Studer et al., 2011), and analysis of teosinte alleles at tb1 has identified functionally distinct allelic classes of tb1 (Studer & Doebley, 2012), but little is known about the role of tb1 or the Hopscotch insertion at this locus in natural populations of teosinte. Previous studies have confirmed the presence of the Hopscotch in samples of parviglumis and landrace maize (Studer et al., 2011); however, little is known about the frequency with which the Hopscotch is segregating in natural populations.

In teosinte and other plants that grow at high population density, individuals detect competition from neighbors via the ratio of red to far-red light. An increase in far-red relative to red light accompanies shading and triggers the shade avoidance syndrome, a suite of physiological and morphological changes such as reduced tillering, increased plant height and early flowering (Kebrom & Brutnell, 2007). The tb1 locus appears to play an important role in the shade avoidance pathway in Zea mays (Lukens & Doebley, 1999) and other grasses (Kebrom & Brutnell, 2007) via changes in expression levels in response to shading. Lukens & Doebley (1999) introgressed the teosinte tb1 allele into a maize inbred background and noted that under low density conditions plants were highly tillered but that under high density, plants showed significantly reduced tillers and grew taller. Based on these results we hypothesize that the Hopscotch (i.e., the domesticated allele) at tb1 may play a role in the ecology of teosinte, especially in high-density populations. In this study we aim to characterize the distribution of the Hopscotch insertion in parviglumis, mexicana, and landrace maize, and to examine the phenotypic effects of the insertion in parviglumis. We use a combination of PCR genotyping for the Hopscotch element in our full panel and sequencing of two small regions upstream of tb1 combined with a larger SNP dataset in a subset of teosinte populations to explore patterns of genetic variation at this locus. Finally, we test for an association between the Hopscotch element and tillering phenotypes in samples from a natural population of parviglumis.

Materials & Methods

Sampling and genotyping

We sampled all individuals and populations that were available to us, consisting of 1,110 individuals from 350 populations (247 maize landraces, 17 mexicana populations, and 86 parviglumis populations) and assessed the presence or absence of the Hopscotch insertion (Table S1 and Table S2). Numbers of individuals sampled per population ranged from 1–43 for parviglumis, 1–35 for mexicana, and 1–18 for the maize landrace populations. Available samples did not allow us to sample evenly from populations, but did allow us to calculate Hopscotch frequency in a subset of populations, as well as elucidate the geographic distribution of the Hopscotch across multiple independent sampling sites. DNA was extracted from leaf tissue using a modified CTAB approach (Doyle & Doyle, 1990; Maloof et al., 1984). We designed primers using PRIMER3 (Rozen & Skaletsky, 2000) implemented in Geneious (Kearse et al., 2012) to amplify the entire Hopscotch element, as well as an internal primer allowing us to simultaneously check for possible PCR bias between presence and absence of the Hopscotch insertion due to its large size (∼5 kb). Two PCRs were performed for each individual, one with primers flanking the Hopscotch (HopF/HopR) and one with a flanking primer (HopF) and an internal primer (HopIntR). Primer sequences are HopF, 5′-TCGTTGATGCTTTGATGGATGG-3′; HopR, 5′-AACAGTATGATTTCATGGGACCG-3′; and HopIntR, 5′-CCTCCACCTCTCATGAGATCC-3′ (Fig. 1 and Fig. S1). Homozygotes for the no-Hopscotch allele show a single band for absence of the element (∼300 bp) produced by the HopF/HopR primer set, and 0 bands for the HopF/HopIntR primer set since they lack the LTR where the internal primer sequence is located. Homozygotes for the Hopscotch allele also show one band at 5 kb for the HopF/HopR PCR product as well as one band at 1.1 kb for the HopF/HopIntR PCR. Heterozygotes for the Hopscotch allele show three bands total; both a 300 bp band and a 5 kb band for the HopF/HopR PCR and a 1.1 Kb band for the HopF/HopIntR PCR (Table S2). Since we developed a PCR protocol for each allele, if only one PCR resolved well, we scored one allele for that individual rather than infer the diploid genotype. We used Phusion High Fidelity Enzyme (Thermo Fisher Scientific Inc., Waltham, Massachusetts, USA) and the following conditions for amplifications: 98 °C for 3 min, 30 cycles of 98 °C for 15 s, 65 °C for 30 s, and 72 °C for 3 min 30 s, with a final extension of 72 °C for 10 min. PCR products were visualized on a 1% agarose gel and scored for presence/absence of the Hopscotch based on band size.

Figure 1 Primer Locations at tb1 Locus.

Representation of the upstream regulatory region of tb1, showing the tb1 coding region (green) and the Hopscotch insertion (red). Arrows show the location of primer sets; in black, primers used for amplification and sequencing (Region 1; within the 5′ UTR, and Region 2; 66,169 bp upstream from the tb1 ORF); in blue, primers used to genotype the Hopscotch insertion. The amplification product for the HopF/HopR is either a 5 kb band (an allele that includes the Hopscotch insertion, or a 300 bp band (an allele that does not include the Hopscotch insertion. The HopF/HopIntR primer combination produces a 1.1 kb band in individuals that have the Hopscotch allele, and no band for individuals that lack the insertion, since the HopIntR primer sits within the LTR

Genotyping analysis

To calculate differentiation between populations (FST) and subspecies (FCT) we used HierFstat (Goudet, 2005). These analyses only included populations (n = 32) in which eight or more chromosomes were sampled. To test the hypothesis that the Hopscotch insertion may be adaptive under certain environmental conditions, we looked for significant associations between Hopscotch frequency and environmental variables using the software BayEnv (Coop et al., 2010). BayEnv creates a covariance matrix of relatedness between populations and then tests a null model that allele frequencies in populations are determined by the covariance matrix of relatedness alone against the alternative model that allele frequencies are determined by a combination of the covariance matrix and an environmental variable, producing a posterior probability (i.e., Bayes Factor; Coop et al., 2010). We used teosinte (ssp. parviglumis and ssp. mexicana) genotyping and covariance data from Pyhäjärvi et al. (2013) for BayEnv, with the Hopscotch insertion coded as an additional biallelic marker. SNP data from Pyhäjärvi et al. (2013) were obtained using the MaizeSNP50 BeadChip and Infinium HD Assay (Illumina, San Diego, CA, USA) and phased using the program fastPHASE (Scheet & Stephens, 2006). Environmental data were previously obtained from www.worldclim.org and soil data were downloaded from the Harmonized World Soil Database (FAO/IIASA/ISRIC/ISSCAS/JRC, 2012) at www.harvestchoice.org. Environmental data represent average values for the last several decades (climatic data) or are likely stable over time (soil data) and therefore represent conditions important for local adaptation of our samples. Information from these data sets was summarized by principle component analysis following Pyhäjärvi et al. (2013).

Sequencing

In addition to genotyping, we chose a subset of parviglumis individuals for sequencing. We chose twelve individuals from each of four populations from Jalisco state, Mexico (San Lorenzo, La Mesa, Ejutla A, and Ejutla B). For amplification and sequencing, we selected two regions approximately 600 bp in size from within the 5′ UTR of tb1 (Region 1) and from 1,235 bp upstream of the start of the Hopscotch (66,169 bp upstream from the start of the tb1 ORF; Region 2). We designed the following primers using PRIMER3 (Rozen & Skaletsky, 2000): for the 5′ UTR, 5′-GGATAATGTGCACCAGGTGT-3′ and 5′-GCGTGCTAGAGACACYTGTTGCT-3′; for the 66 kb upstream region, 5′-TGTCCTCGCCGCAACTC-3′ and 5′-TGTACGCCCGCCCCTCATCA-3′ (Table S1, See Supplemental Materials with the online version of this article). We used Taq polymerase (New England Biolabs Inc., Ipswich, Massachusetts, USA) and the following thermal cycler conditions to amplify fragments: 94 °C for 3 min, 30 cycles of 92 °C for 40 s, annealing for 1 min, 72 °C for 40 s, and a final 10 min extension at 72 °C. Annealing temperatures for Region 1 and Region 2 were 59.7 °C and 58.8 °C, respectively. To clean excess primer and dNTPs we added two units of Exonuclease1 and 2.5 units of Antarctic Phosphatase to 8.0 µL of amplification product. This mix was placed on a thermal cycler with the following program: 37 °C for 30 min, 80 °C for 15 min, and a final cool-down step to 4 °C.

We cloned cleaned fragments into a TOPO-TA vector (Life Technologies, Grand Island, New York, USA) using OneShot TOP10 chemically competent E. coli cells, with an extended ligation time of 30 min for a complex target fragment. We plated cells on LB agar plates containing kanamycin, and screened colonies using vector primers M13 Forward and M13 Reverse under the following conditions: 96 °C for 5 min; then 35 cycles at 96 °C for 30 s, 53 °C for 30 s, 72 °C for 2 min; and a final extension at 72 °C for 4 min. We visualized amplification products for incorporation of our insert on a 1% agarose TAE gel.

Amplification products with successful incorporation of our insert were cleaned using Exonuclease 1 and Antarctic Phosphatase following the procedures detailed above, and sequenced with vector primers M13 Forward and M13 Reverse using Sanger sequencing at the College of Agriculture and Environmental Sciences (CAES) sequencing center at UC Davis. We aligned and trimmed primer sequences from resulting sequences using the software Geneious (Kearse et al., 2012). Following alignment, we verified singleton SNPs by sequencing an additional one to four colonies from each clone. If the singleton was not present in these additional sequences it was considered an amplification or cloning error, and we replaced the base with the base of the additional sequences. If the singleton appeared in at least one of the additional sequences we considered it a real variant and kept it for further analyses.

Sequence analysis

For population genetic analyses of sequenced Region 1 and sequenced Region 2 we used the Analysis package from the Libsequence library (Thornton, 2003) to calculate pairwise FST between populations and to calculate standard diversity statistics (number of haplotypes, haplotype diversity, Watterson’s estimator θˆW, pairwise nucleotide diversity θˆπ, and Tajima’s D). Significance of Tajima’s D results was gauged by comparing empirical data to 10,000 coalescent simulations conducted using the program ms (Hudson, 2002) under a standard neutral model based on observed estimates of the population mutation rate theta and assuming an identical value for the population recombination rate rho. Empirical results falling outside the 95% confidence interval of our simulated data were deemed significant. To produce a visual representation of differentiation between sequences and examine patterns in sequence clustering by Hopscotch genotype, we used Phylip (http://evolution.genetics.washington.edu/phylip.html) to create neighbor-joining trees with bootstrap-supported nodes (10,000 repetitions). For creation of trees we also included homologous sequence data from Maize HapMapV2 (Chia et al., 2012) for teosinte inbred lines (TILs), some of which are known to be homozygous for the Hopscotch insertion (TIL03, TIL17, TIL09), as well as 59 lines of domesticated maize.

Introgression analysis

In order to assess patterns of linkage disequilibrium (LD) around the Hopscotch element in the context of chromosomal patterns of LD we used Tassel (Bradbury et al., 2007) and calculated LD between SNPs across chromosome 1 using previously published data from twelve plants each of the Ejutla A (EjuA), Ejutla B (EjuB), San Lorenzo (SLO), and La Mesa (MSA) populations (Pyhäjärvi et al., 2013). We chose these populations because we had both genotyping data for the Hopscotch as well as chromosome-wide SNP data for chromosome 1. For each population, we filtered the initial set of 5,897 SNPs on chromosome 1 to accept only SNPs with a minor allele frequency of at least 0.1, resulting in 1,671, 3,023, 3,122, and 2,167 SNPs for SLO, EjuB, EjuA, and MSA, respectively. We then used Tassel (Bradbury et al., 2007) to calculate linkage disequilibrium (r2) across chromosome 1 for each population.

We examined evidence of introgression on chromosome 1 in these same four populations (EjuA, EjuB, MSA, SLO) using STRUCTURE (Falush, Stephens & Pritchard, 2003) and phased data from Pyhäjärvi et al. (2013), combined with the corresponding SNP data from a diverse panel of 282 maize lines (Cook et al., 2012). SNPs were anchored in a modified version of the IBM genetic map (Gerke et al., 2013). Since STRUCTURE does not account for LD due to physical linkage we created haplotype blocks using a custom Perl script from Hufford et al. (2013), code available at http://dx.doi.org/10.6084/m9.figshare.1165577. In maize, LD decays over an average distance of 5,500 bp (Chia et al., 2012); because LD decay is even more rapid in teosinte (Chia et al., 2012) we used a conservative haplotype block size of 5 kb. We ran STRUCTURE at K = 2 under the linkage model, with the assumption being that individuals fall into either a maize or teosinte cluster, performing three replicates with an MCMC burn-in of 10,000 steps and 50,000 steps post burn-in.

Phenotyping of parviglumis

To investigate the phenotypic effects of the Hopscotch insertion in teosinte we conducted a phenotyping trial in which we germinated 250 seeds of parviglumis collected in Jalisco state, Mexico (population San Lorenzo; Hufford, 2010) where the Hopscotch insertion is segregating at highest frequency (0.44) in our initial genotyping sample set. In order to increase our chances of finding the Hopscotch in our association population we selected seeds from sites within the population where genotyped individuals were homozygous or heterozygous for the insertion. We chose between 10–13 seeds from each of 23 sampling sites. We treated seeds with Captan fungicide (Southern Agricultural Insecticides Inc., Palmetto, Florida, USA) and germinated them in petri dishes with filter paper. Following germination, 206 successful germinations were planted into one-gallon pots with potting soil and randomly spaced one foot apart on greenhouse benches. Plants were watered three times a day with an automatic drip containing 10-20-10 fertilizer, which was supplemented with hand watering on extremely hot and dry days.

Starting on day 15, we measured tillering index as the ratio of the sum of tiller lengths to the height of the plant (Briggs et al., 2007). Following initial measurements, we phenotyped plants for tillering index every 5 days through day 40, and then on day 50 and day 60. On day 65 we measured culm diameter between the third and fourth nodes of each plant. Following phenotyping we extracted DNA from all plants using a modified SDS extraction protocol. We genotyped individuals for the Hopscotch insertion following the PCR protocols listed above.

Tillering index data for each genotypic class did not meet the criteria for a repeated measures ANOVA, so we transformed the data with a Box–Cox transformation (λ = 0) in the Car Package for R (Fox & Weisberg, 2011) to improve the normality and homogeneity of variance among genotype classes. We analyzed relationships between genotype and tillering index and tiller number using a repeated measures ANOVA through a general linear model function implemented in SAS v.9.3 (SAS Institute Inc., Cary, North Carolina, USA). Additionally, in order to compare any association between Hopscotch genotype and tillering and associations at other presumably unrelated traits, we performed an ANOVA between culm diameter and genotype using the same general linear model in SAS. Culm diameter is not believed to be correlated with tillering index or variation at tb1 and is used as our independent trait for phenotyping analyses. SAS code used for analysis is available at http://dx.doi.org/10.6084/m9.figshare.1166630.

Results

Genotyping for the Hopscotch insertion

The genotype at the Hopscotch insertion was confirmed with two PCRs for 837 individuals of the 1,100 screened (Table S1 and Table S2). Among the 247 maize landrace accessions genotyped, all but eight were homozygous for the presence of the insertion. Within our parviglumis and mexicana samples we found the Hopscotch insertion segregating in 37 (n = 86) and four (n = 17) populations, respectively, and at highest frequency within populations in the states of Jalisco, Colima, and Michoacán in central-western Mexico (Fig. 2). Using our Hopscotch genotyping, we calculated differentiation between populations (FST) and subspecies (FCT) for populations in which we sampled sixteen or more chromosomes. We found that FCT = 0, and levels of FST among populations within each subspecies (0.22) and among all populations (0.23) (Table 1) are similar to genome-wide estimates from previous studies Pyhäjärvi et al., 2013. Although we found large variation in Hopscotch allele frequency among our populations, BayEnv analysis did not indicate a correlation between the Hopscotch insertion and environmental variables (all Bayes Factors <1).

Figure 2 Map of parviglumis populations and Hopscotch allele frequency.

Map showing the frequency of the Hopscotch allele in populations of parviglumis where we sampled more than 6 individuals. Size of circles reflects number of individuals sampled. The Balsas River is shown, as the Balsas River Basin is believed to be the center of domestication of maize.

Table 1 Pairwise FST values.

Pairwise FST values from sequence and Hopscotch genotyping data.

Comparison	Region 1	Region 2	Hopscotch	
EjuA & EjuB	0	0	0	
EjuA & MSA	0.326	0.328	0.186	
EjuA & SLO	0.416	0.258	0.280	
EjuB & MSA	0.397	0.365	0.188	
EjuB & SLO	0.512	0.290	0.280	
MSA & SLO	0.007	0	0.016	

Sequencing upstream regions of the tb1 ORF

To investigate patterns of sequence diversity and linkage disequilibrium (LD) in the tb1 region and any evidence of selection on this locus, we sequenced two small (<1 kb) regions upstream of the tb1 ORF in four populations from the Jalisco region. After alignment and singleton checking we recovered 48 and 40 segregating sites for the 5′ UTR region (Region 1) and the 66 kb upstream region (Region 2), respectively. For Region 1, Ejutla A has the highest values of haplotype diversity and θˆπ, while Ejutla B and La Mesa have comparable values of these summary statistics, and San Lorenzo has much lower values. Additionally, Tajima’s D is significantly negative in the two Ejutla populations and La Mesa, but is closer to zero in San Lorenzo (Table 2). For Region 2, haplotype diversity and θˆπ, are similar for Ejutla A and Ejutla B, while La Mesa and San Lorenzo have slightly lower values for these statistics (Table 2). Tajima’s D is positive in all populations except La Mesa, where a slightly negative value suggests a slight excess of low frequency variants (Table 2). Pairwise values of FST within population pairs Ejutla A/Ejutla B and San Lorenzo/La Mesa are close to zero for both sequenced regions as well as for the Hopscotch, while they are high for other population pairs (Table 1).

Table 2 Population genetic statistics.

Population genetic statistics from resequenced regions near the tb1 locus. Significant values are marked with an asterisk.

Population	# Haplotypes	Hap. Diversity	θˆπ	Tajima’s D	
Region 1(5′ UTR)	
EJUA	8	0.859	0.005	−1.650*	
EJUB	5	0.709	0.004	−1.831*	
MSA	6	0.682	0.004	−1.755*	
SLO	3	0.318	0.001	−0.729	
Region 2 (66 kb upstream)	
EJUA	8	0.894	0.018	0.623	
EJUB	8	0.894	0.016	0.295	
MSA	3	0.682	0.011	−0.222	
SLO	4	0.742	0.014	0.932	

Evidence of introgression around the tb1 region

We investigated the possibility of introgression as an explanation for the frequency of the Hopscotch allele in populations of teosinte using previously collected SNP data from Pyhäjärvi et al. (2013). The highest frequency of the Hopscotch insertion in teosinte was found in parviglumis sympatric with cultivated maize. Our initial hypothesis was that the high frequency of the Hopscotch element in these populations could be attributed to introgression from maize into teosinte. To investigate this possibility, we examined overall patterns of linkage disequilibrium across chromosome 1 and specifically in the tb1 region. If the Hopscotch is found in these populations due to recent introgression from maize, we would expect to find large blocks of linked markers near this element. We find no evidence of elevated linkage disequilibrium between the Hopscotch and SNPs surrounding the tb1 region in our resequenced populations (Fig. 3), and r2 in the tb1 region does not differ significantly between populations with (average r2 of 0.085) and without (average r2 = 0.082) the Hopscotch insertion. In fact, average r2 is lower in the tb1 region (r2 = 0.056) than across the rest of chromosome 1 (r2 = 0.083; Table 3).

Figure 3 Linkage Disequilibrium along Chromosome 1.

Linkage disequilibrium for SNPs in Mb 261–268 on chromosome 1. The yellow rectangle indicates the location of the Hopscotch insertion and the green rectangle represents the tb1 ORF. (A) Ejutla A; (B) Ejutla B; (C) La Mesa; (D) San Lorenzo. The upper triangle above the black diagonal is colored based on the r2 value between SNPs while the bottom triangle is colored based on p-value for the corresponding r2 value.

Table 3 Chromosome-wide r2 values.

Mean r2 values between SNPs on chromosome 1, in the broad tb1 region, within the 5′ UTR of tb1 (Region 1), and 66 kb upstream of tb1 (Region 2).

Population	Chr. 1	tb1 region	Region 1	Region 2	
Ejutla A	0.095	0.050	0.747	0.215	
Ejutla B	0.069	0.051	0.660	0.186	
La Mesa	0.070	0.053	0.914	0.766	
San Lorenzo	0.101	0.067	0.912	0.636	

Neighbor joining trees of our sequence data and data from the teosinte inbred lines (TILs; data from Maize HapMapV2, Chia et al., 2012) do not reveal any clear clustering pattern with respect to population or Hopscotch genotype (Fig. S2, see Supplemental Information with the online version of this article); individuals within our sample that have the Hopscotch insertion do not group with the teosinte inbred lines or domesticated maize that have the Hopscotch insertion. The lack of clustering of Hopscotch genotypes in our NJ tree as well as the lack of LD around tb1 do not support the hypothesis that the Hopscotch insertion in these populations of parviglumis is the result of recent introgression. However, to further explore this hypothesis we performed a STRUCTURE analysis using Illumina MaizeSNP50 data from four of our parviglumis populations (EjuA, EjuB, MSA, and SLO) (Pyhäjärvi et al., 2013) and the maize 282 diversity panel (Cook et al., 2012). The linkage model implemented in STRUCTURE can be used to identify ancestry of blocks of linked variants which would arise as the result of recent admixture between populations. If the Hopscotch insertion is present in populations of parviglumis as a result of recent admixture with domesticated maize, we would expect the insertion and linked variants in surrounding sites to be assigned to the “maize” cluster in our STRUCTURE runs, not the “teosinte” cluster. In all runs, assignment to maize in the tb1 region across all four parviglumis populations is low (average 0.017) and much below the chromosome-wide average (0.20; Table 4 and Fig. 4).

Figure 4 STRUCTURE assignment to Maize near tb1.

STRUCTURE assignment to maize across a section of chromosome 1. The dotted lines mark the beginning of the sequenced region 66 kb upstream (Region 2) and the end of the tb1 ORF.

Table 4 STRUCTURE assignment near tb1.

Assignments to maize and teosinte in the tb1 and chromosome 1 regions from STRUCTURE.

	tb1 region	Chr 1	
Population	Maize	Teosinte	Maize	Teosinte	
Ejutla A	0.022	0.978	0.203	0.797	
Ejutla B	0.019	0.981	0.187	0.813	
La Mesa	0.012	0.988	0.193	0.807	
San Lorenzo	0.016	0.984	0.205	0.795	

Phenotyping of Zea mays ssp. parviglumis

To assess the contribution of tb1 to phenotypic variation in tillering in a natural population, we grew plants from seed sampled from the San Lorenzo population of parviglumis, which had a high mean frequency (0.44) of the Hopscotch insertion based on our initial genotyping. We measured tiller number and tillering index, the ratio of the sum of tiller lengths to plant height, for 206 plants from within the San Lorenzo population, and genotyped plants for the Hopscotch insertion. We also measured culm diameter, a phenotype that differs between maize and teosinte but has not been shown to be affected by the Hopscotch insertion (Briggs et al., 2007). Culm diameter is meant to be an independent trait against which we can compare patterns of tillering index x Hopscotch genotype data. If tillering index in parviglumis is affected by the Hopscotch insertion, the expectation is that patterns of tillering index data will have a significant correlation with Hopscotch genotype, whereas we should find no significant correlation between culm diameter and Hopscotch genotype. Phenotypic data are available at http://dx.doi.org/10.6084/m9.figshare.776926. Our plantings produced 82 homozygotes for the Hopscotch insertion at tb1, 104 heterozygotes, and 20 homozygotes lacking the insertion; these numbers do not deviate from expectations of Hardy-Weinberg equilibrium. After performing a repeated measures ANOVA between our transformed tillering index data and Hopscotch genotype, we find no significant correlation between genotype at the Hopscotch insertion and tillering index (Fig. 5), tiller number, or culm diameter. Only on day 40 did we observe a weak but statistically insignificant (r2 = 0.02, p = 0.0848) correlation between tillering index and the Hopscotch genotype, although in the opposite direction of that expected, with homozygotes for the insertion showing a higher tillering index.

Figure 5 Tillering Index in parviglumis.

Box-plots showing tillering index in greenhouse grow-outs of parviglumis for phenotyping. White indicates individuals homozygous for the Hopscotch, light grey represents heterozygotes, and dark grey represents homozygotes for the teosinte (No Hopscotch) allele. Within boxes, dark black lines represent the median, and the edges of the boxes are the first and third quartiles. Outliers are displayed as dots, the maximum value excluding outliers is shown with the top whisker, while the minimum excluding outliers is shown with the bottom whisker.

Discussion

Adaptation occurs due to selection on standing variation or de novo mutations. Adaptation from standing variation has been well-described in a number of systems; for example, selection for lactose tolerance in humans (Plantinga et al., 2012; Tishkoff et al., 2007), variation at the Eda locus in three-spined stickleback (Kitano et al., 2008; Colosimo et al., 2005), and pupal diapause in the Apple Maggot fly (Feder et al., 2003). Although the adaptive role of standing variation has been described in many systems, its importance in domestication is not as well studied.

In maize, alleles at domestication loci (RAMOSA1, Sigmon & Vollbrecht, 2010; barren stalk1, Gallavotti et al., 2004; and grassy tillers1, Whipple et al., 2011) are thought to have been selected from standing variation, suggesting that diversity already present in teosinte may have played an important role in maize domestication. The teosinte branched1 gene is one of the best characterized domestication loci, and, while previous studies have suggested that differences in plant architecture between maize and teosinte are a result of selection on standing variation at this locus (Clark et al., 2006; Studer et al., 2011), much remains to be discovered regarding natural variation at this locus and its ecological role in teosinte.

Studer et al. (2011) genotyped 90 accessions of teosinte (inbred and outbred), providing the first evidence that the Hopscotch insertion is segregating in teosinte. Given that the Hopscotch insertion has been estimated to predate the domestication of maize, it is not surprising that it can be found segregating in populations of teosinte. However, by widely sampling across teosinte populations our study provides greater insight into the distribution and prevalence of the Hopscotch in teosinte. While our findings are consistent with Studer et al. (2011) in that we identify the Hopscotch allele segregating in teosinte, we find it at higher frequency than previously suggested. Moreover, many of our parviglumis populations with a high frequency of the Hopscotch allele fall in the Jalisco cluster identified by Fukunaga et al. (2005), and further distinguish this region from the Balsas River Basin where maize was domesticated (Matsuoka et al., 2002). Potential explanations for the high frequency of the Hopscotch element in parviglumis from the Jalisco cluster include gene flow from maize, genetic drift, and natural selection.

While gene flow from crops into their wild relatives is well-known, (Ellstrand, Prentice & Hancock, 1999; Zhang et al., 2009; Thurber et al., 2010; Baack et al., 2008; Hubner et al., 2012; Wilkes, 1977; van Heerwaarden et al., 2011; Barrett, 1983), our results do not suggest introgression from maize at the tb1 locus, and are more consistent with Hufford et al. (2013) who found resistance to introgression from maize into mexicana around domestication loci. Clustering in our NJ trees does not reflect the pattern expected if maize alleles at the tb1 locus had introgressed into populations of teosinte. Moreover, there is no signature of elevated LD in the tb1 region relative to the rest of chromosome 1, and Bayesian assignment to a maize cluster in this region is both low and below the chromosome-wide average (Fig. 4, Table 4). Together, these data point to an explanation other than recent introgression for the high observed frequency of Hopscotch in the Jalisco cluster of our parviglumis populations.

Although recent introgression seems unlikely, we cannot rule out ancient introgression as an explanation for the presence of the Hopscotch in these populations. If the Hopscotch allele was introgressed in the distant past, recombination may have broken up LD, a process that would be consistent with our data. We find this scenario less plausible, however, as there is no reason why gene flow should have been high in the past but absent in present-day sympatric populations. In fact, early generation maize-teosinte hybrids are common in these populations (MB Hufford, pers. obs., 2010), and genetic data support ongoing gene flow between domesticated maize and both mexicana and parviglumis in a number of sympatric populations (Hufford et al., 2013; Ellstrand et al., 2007; van Heerwaarden et al., 2011; Warburton et al., 2011).

Remaining explanations for differential frequencies of the Hopscotch among teosinte populations include both genetic drift and natural selection. Previous studies using both SSRs and genome-wide SNP data have found evidence for a population bottleneck in the San Lorenzo population (Hufford, 2010; Pyhäjärvi et al., 2013), and the lower levels of sequence diversity in this population in the 5′ UTR (Region 1) coupled with more positive values of Tajima’s D are consistent with these earlier findings. Such population bottlenecks can exaggerate the effects of genetic drift through which the Hopscotch allele may have risen to high frequency entirely by chance. A bottleneck in San Lorenzo, however, does not explain the high frequency of the Hopscotch in multiple populations in the Jalisco cluster. Moreover, available information on diversity and population structure among Jaliscan populations (Hufford, 2010; Pyhäjärvi et al., 2013) is not suggestive of recent colonization or other demographic events that would predict a high frequency of the allele across populations. Finally, diversity values in the 5′ UTR of tb1 are suggestive of natural selection acting upon the gene in populations of parviglumis. Overall nucleotide diversity is 76% less than seen in the sequences from the 66 kb upstream region, and Tajima’s D is considerably lower and consistently negative across populations (Table 2). In fact, values of Tajima’s D in the 5′ UTR are toward the extreme negative end of the distribution of this statistic previously calculated across loci sequenced in parviglumis (Wright et al., 2005; Moeller, Tenaillon & Tiffin, 2007) and significantly negative in three of our surveyed populations (EjuA, EjuB, MSA) based on coalescent simulations under a standard neutral model. Though not definitive, these results are consistent with the action of selection on the upstream region of tb1, perhaps suggesting an ecological role for the gene in Jaliscan populations of parviglumis. Finally, while these results are consistent with selection at the tb1 locus in teosinte, they do not confirm selection specifically on the Hopscotch insertion at this locus.

Significant effects of the Hopscotch insertion on lateral branch length, number of cupules, and tillering index in domesticated maize have been well documented (Studer et al., 2011). Weber et al. (2007) described significant phenotypic associations between markers in and around tb1 and lateral branch length and female ear length in a sample from 74 natural populations of parviglumis (Weber et al., 2007); however, these data did not include markers from the Hopscotch region 66 kb upstream of tb1. Our study is the first to explicitly examine the phenotypic effects of the Hopscotch insertion across a wide collection of individuals sampled from natural populations of teosinte. We have found no significant effect of the Hopscotch insertion on tillering index or tiller number, a result that is discordant with its clear phenotypic effects in maize. It is possible that the planting density of our seedlings (plants spaced 12 inches apart) was too high, leading to an overall decrease in tillering as previously seen in Lukens & Doebley (1999). This factor may have limited our capacity to observe variation in tillering index.

An alternative interpretation of this result would be that the Hopscotch controls tillering in maize (Studer et al., 2011), but tillering in teosinte is affected by variation at other loci. Consistent with this interpretation, tb1 is thought to be part of a complex pathway controlling branching, tillering and other phenotypic traits (Kebrom & Brutnell, 2007; Clark et al., 2006).

A recent study by Studer & Doebley (2012) examined variation across traits in an allelic series study of the tb1 locus. Studer & Doebley (2012) introgressed nine unique teosinte tb1 segments (one from Zea diploperennis, and four each from mexicana and parviglumis) into an inbred maize (W22) background and investigated their phenotypic effects. Their findings suggest that different teosinte tb1 segments produce equivalent effects on tillering and that variation in tillering observed across these taxa is not due to a tb1 allelic series but potentially due to variation at other, unlinked loci. Clues to the identity of these loci may be found in QTL studies that have identified loci controlling branching architecture (e.g., Doebley & Stec, 1991; Doebley & Stec, 1993). Many of these loci (grassy tillers, gt1; tassel-replaces-upper-ears1, tru1; terminal ear1, te1) have been shown to interact with tb1 (Whipple et al., 2011; Li, 2012), and both tru1 and te1 affect the same phenotypic traits as tb1 (Doebley, Stec & Gustus, 1995). tru1, for example, has been shown to act either epistatically or downstream of tb1, affecting both branching architecture (decreased apical dominance) and tassel phenotypes (shortened tassel and shank length and reduced tassel number; Li, 2012). Variation in these additional loci may have affected tillering in our collections and contributed to the lack of correlation we see between Hopscotch genotype and tillering.

Conclusions

In conclusion, our findings demonstrate that the Hopscotch allele is widespread in populations of parviglumis and mexicana and occasionally at high allele frequencies. Analysis of linkage using SNPs from across chromosome 1 does not suggest that the Hopscotch allele is present in these populations due to recent introgression, and it seems unlikely that the insertion would have drifted to high frequency in multiple populations. We do, however, find preliminary evidence of selection on the tb1 locus in parviglumis. Coupled with our observation of high frequency of the Hopscotch insertion in a number of populations, this suggests that the locus—and potentially the domestication allele at this locus—may play an ecological role in teosinte.

In contrast to domesticated maize, the Hopscotch insertion does not appear to have a large effect on tillering in a diverse sample of parviglumis from a natural population and the phenotypic consequences of variation at tb1 thus remain unclear. Future studies should examine expression levels of tb1 in teosinte with and without the Hopscotch insertion and further characterize the effects of additional loci involved in branching architecture (e.g., gt1, tru1, and te1). These data, in conjunction with more exhaustive phenotyping, should help to further clarify the ecological significance of the domesticated tb1 allele in natural populations of teosinte.

Supplemental Information

Table S1 Genotyped Teosinte Accessions

Accessions of Zea mays ssp. mexicana (RIMME) and Zea mays ssp. parviglumis (RIMPA) sampled. RIHY is a Z. mays ssp. parviglumis and Zea mays ssp. mays hybrid.

Click here for additional data file.

Table S2 Hopscotch Frequency in Maize

Hopscotch frequency in sampled Zea mays ssp. mays (RIMMA).

Click here for additional data file.

Figure S1 Gel Image of PCR Assays for Hopscotch Allele

Agarose gel image of amplification products for genotyping of the Hopscotch element. Lanes 1 (HopF/HopR; 5 kb band) and 2 (HopF/HopIntR; 1.1 kb) are the products for one individual that is homozygous for the element; Lanes 3 (HopF/HopR; 5 kb band) and 4 (HopF/HopIntR; 1.1 kb) are also the products of an individual that is homozygous for the element; and Lanes 5 (HopF/HopR; 300 bp) and 6 (HopF/HopIntR; N/A) are the products of an individual that is homozygous for the teosinte (lacking the Hopscotch) allele.

Click here for additional data file.

Figure S2 Neighbor-Joining Trees Based on Sequenced Regions

Neighbor-joining tree of the sequenced region in the 5′ UTR (right; Region 1) and the 66,169 bp upstream region (left; Region 2) of tb1 using 10,000 bootstraps. Individuals with genotype data are colored: Homozygous for the teosinte (no Hopscotch) allele (red), homozygous for the maize (Hopscotch) allele (blue), heterozygotes (purple). TILs (teosinte inbred lines) are colored in green, with stars indicating the 3 TILs known to have the Hopscotch insertion. Black indicates individuals not genotyped for the Hopscotch insertion. EjuA refers to individuals from population Ejutla A, EjuB from Ejutla B, SLO from San Lorenzo, and MSA from La Mesa. Remaining individuals are lines of maize (Zea mays ssp. mays).

Click here for additional data file.

The authors thank Graham Coop for helpful discussion and Lauryn Brown, Joshua Hegarty, Pui Yan Ho, and Garry Pearson for assistance with the phenotyping portion of this study.

Additional Information and Declarations

Competing Interests

Author Contributions

DNA Deposition

Data Deposition

Jeffrey Ross-Ibarra is an Academic Editor for PeerJ.

Laura Vann performed the experiments, analyzed the data, wrote the paper, prepared figures and/or tables, reviewed drafts of the paper.

Thomas Kono performed the experiments, analyzed the data, reviewed drafts of the paper.

Tanja Pyhäjärvi analyzed the data, contributed reagents/materials/analysis tools, reviewed drafts of the paper.

Matthew B. Hufford conceived and designed the experiments, performed the experiments, contributed reagents/materials/analysis tools, wrote the paper, prepared figures and/or tables, reviewed drafts of the paper.

Jeffrey Ross-Ibarra conceived and designed the experiments, performed the experiments, contributed reagents/materials/analysis tools, wrote the paper, reviewed drafts of the paper.

The following information was supplied regarding the deposition of DNA sequences:

Figshare http://dx.doi.org/10.6084/m9.figshare.1352094, http://dx.doi.org/10.6084/m9.figshare.1352093.

The following information was supplied regarding the deposition of related data:

Figshare: http://dx.doi.org/10.6084/m9.figshare.1166630, http://dx.doi.org/10.6084/m9.figshare.776926, http://dx.doi.org/10.6084/m9.figshare.1165577, http://dx.doi.org/10.6084/m9.figshare.779707.

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
