# Peer review of "Natural variation in teosinte at the domestication locus teosinte branched1 (tb1)"

_PeerJ, doi:10.7717/peerj.900_

## Round 0.1 · original submission · Minor Revisions

The reviewers comments seems to be largely in agreement and self-explanatory. Please be sure to address the comment regarding the validity of the PCR assays raised by Reviewer 2.

Reviewer 1 ·

Basic reporting

The submission must adhere to all PeerJ policies:

--To my knowledge, yes.

The article must be written in English using clear and unambiguous text and must conform to professional standards of courtesy and expression:

--Well written and organized.

The article should include sufficient introduction and background to demonstrate how the work fits into the broader field of knowledge. Relevant prior literature should be appropriately referenced:

--The hypotheses and findings are appropriately presented and discussed in the context of the prior literature.

The structure of the submitted article should conform to one of the templates:

-- The editorial office should be responsible for checking this, not the peer reviewers.

Figures should be relevant to the content of the article, of sufficient resolution, and appropriately described and labeled:

-- I have recommended some modifications. See specific comments below.

The submission should be ‘self-contained,’ should represent an appropriate ‘unit of publication’, and should include all results relevant to the hypothesis.

-- All OK.

Experimental design

The submission must describe original primary research within the Scope of the journal:

--YES

The submission should clearly define the research question, which must be relevant and meaningful:

--YES

The investigation must have been conducted rigorously and to a high technical standard:

--YES. See below for some quibbles about statistical significance testing.

Methods should be described with sufficient information to be reproducible by another investigator:

-- See below for request for clarification on one analysis.

The research must have been conducted in conformity with the prevailing ethical standards in the field:

--YES

Validity of the findings

The data should be robust, statistically sound, and controlled:

--YES

The data on which the conclusions are based must be provided or made available in an acceptable discipline-specific repository:

--YES. I haven't checked all online links cited in the paper.

The conclusions should be appropriately stated, should be connected to the original question investigated, and should be limited to those supported by the results:

--OK

Speculation is welcomed, but should be identified as such:

--OK

Decisions are not made based on any subjective determination of impact, degree of advance, novelty, being of interest to only a niche audience, etc. Replication experiments are encouraged (provided the rationale for the replication is clearly described), however we do not allow the ‘pointless’ repetition of well known, widely accepted results:

--OK

Additional comments

Populations of crop wild relatives often carry alleles associated with domestication traits at low to moderate frequencies. The presence of these alleles could potentially reflect post-domestication crop-to-wild gene flow or standing genetic variation that predates the domestication process. This study examines the frequency and phenotypic consequences of one of the best characterized crop domestication alleles, maize tb1, in populations of its wild ancestor, teosinte. The tb1 gene functions as a shade avoidance-mediated repressor of organ growth. In domesticated maize, a Hopscotch TE insertion in the tb1 promoter leads to enhanced gene expression and the loss of tillering and branching that characterizes domesticated maize. In this paper, the authors use PCR assays to survey for the presence of the TE insertion in a large sample of accessions, including maize, its wild ancestor teosinte, and highland teosinte (a separate subspecies less closely related to the domesticate). The study examines the potential ecological significance of the TE insertion allele through a Bayenv analysis, and population genetic analyses are used to confirm previous findings that the TE insertion predates maize domestication.

The paper is generally well written, and the conclusions are basically sound. Addressing the points below would improve the paper.

1) The terminology on “accessions,” “individuals,” and “populations” is confusing (e.g., 67-71). Please clarify the difference between individuals and accessions (e.g., line 70 — is this referring to the number of individuals per accession? How are individuals of one accession related? Are these full sibs from the same maternal plant?..)

2) Is the “Locality” designation in Table S1 the same as “population” as used in the text? Please clarify and/or be consistent with this terminology.

3) Fig. 1 should indicate the boundaries of the Mexican states that are used in describing population locations in the text (Jalisco, Colima, etc.). The four focal populations from Jalisco state should also be labeled in this figure (i.e., San Lorenzo, La Mesa, Ejutla A, Ejutla B).

4) It would be helpful to include Fig. S1 in the main text of the paper and to label the two sequenced regions (Region 1 and 2) in the figure itself (not just in the caption).

5) Fig. S2. Annotate this figure to show the size (bp) of each band in the 1 kb ladder. The caption should also define the abbreviations used in the labels.

6) What does “No Hop/Pif” indicate in Fig. S2? It doesn’t have the three predicted bands of a TE heterozygote, and lane 6 seems to be empty altogether. Please clarify.

7) Line 118. Figure S1 should be cited here, not Table S1.

8) Line 150. Why were only 100 bootstrap replicates used for the NJ analysis? 1000 is more typical.

9) Lines 164-175. It would be helpful to provide some more detail on the methods used in defining haplotypes in the STRUCTURE introgression analysis. If I understand correctly, SNPs within 5 kb windows were used to define haplotypes for that window. Were the haplotypes then used to define diploid genotypes for each individual in the STRUCTURE input file? If so, how was phasing performed in defining the haplotypes? Is the output in Fig. 3 showing maize population assignment (i.e., membership coefficient values) for adjacent 5 kb windows? Was the STRUCTURE analysis performed only for the 8 Mb region shown in Fig. 3, or do the results presented in Table 4 reflect the entire chromosome?

11) Line 186. How does the 1-foot spacing of plants in the phenotyping experiment compare to conditions used in the experiments of Lukens and Doebley that documented density-dependent tillering? It could be worth bringing this up in the discussion of the negative phenotyping results (i.e., Fig 4 and Discussion, p. 13).

12) Line 223. Indicate here that these four populations are parviglumis and are from Jalisco state.

13) Line 228. Why is Table S2 cited here? That table doesn’t include any information on nucleotide diversity estimators.

14) Table 2 and the corresponding text (p. 9) should indicate which Tajima’s D values, if any, are statistically significant. If none are statistically significant, then this should be pointed out in the text.

15) Lines 231-232. Suggested rewording: “in all populations except La Mesa, where a slightly negative value suggests a slight excess of low frequency variants…”

16) Lines 249-250. “average r2 is slightly lower in the tb1 region..” Also, include these mean values as a row in Table 3.

17) Fig. 3 could be moved to online supplementary data, since the key information with respect to the tb1 region is already presented in Table 4.

18) Discussion, lines 306-8. “The Hopscotch allele is more prevalent in parviglumis than in mexicana in our sample, suggesting a different history of the allele amongst teosinte subspecies.” This inference is contradicted by the complete lack of differentiation between the subspecies for the TE (Fct = 0; Results, lines 213-217). This should be addressed. I don’t see any evidence presented that the TE differences between the subspecies are statistically significant. Given the unbalanced geographical sampling of the two subspecies (see Table S1), I would also question whether these samples are appropriate for comparing TE frequencies in this way.

19) There are some sloppy inferences about selection in the Discussion and Conclusion — e.g., lines 345-354, 388-389: there’s no evidence that the Tajima’s D values cited here are actually statistically significant. The statement and citations in lines 349-351 suggests that they likely are, but this should be backed up with tests of statistical significance. See comment 14 above.

Reviewer 2 ·

Basic reporting

In the Introduction, the authors give some background about teosinte, but do so mainly as the precursor of maize, not so much as a wild species. For instance, they do not give much information about the ecological differences between parviglumis and mexicana, or about what is known for this species in terms of population structure, population density or ecological behavior. This should be added. In particular, the authors state that the tb1 gene is suspected to play a role in shade avoidance, so information on population density should be highlighted: Is it similar in all teosinte populations? Does it vary with altitude? With other ecological data? Does it vary between parviglumis and mexicana?
The authors refer several times to a previous study they performed: Pyhäjärvi et al., 2013. This is fine, but it is not always easy to understand which results are new and which come from this previous study. For instance, on line 158, it is stated “we had” while whole genome SNP data come from Pyhäjärvi et al., 2013. Similarly, “Table 1” on line 217 should be moved right after “populations (0.23)” on line 216 to avoid confusion.
In the Results section, some titles like “Genotyping”, “Sequencing” or “Phenotyping” are too vague, and sound more like Materials and Methods than to Results. They should be modified in order to highlight what question is addressed.
Paragraph “Neighbor joining trees… insertion”” on lines 235-239 should be moved to the section “Evidence of introgression”.

Experimental design

The biological question is to characterize whether the Hopscotch transposable element at tb1 (which enhances expression of tb1) plays a role in the ecology of teosinte, especially in high-density populations. To do so, they characterize the distribution of this element in parviglumis, mexicana and maize landraces and they examine the phenotypic effects of the insertion in parviglumis.
The authors state that they sampled 1,110 individuals from 350 accessions, with between 1-18 and 1-43 individuals per population. However, they do not provide explanation on (i) how these populations were chosen and (ii) why a different number of individual was sampled per accession (visible also in suppl. Tables S1 and S2). This should be explained.
In Suppl. Table S1, some of the parviglumis accessions are listed as “Breeders line”. Can these be considered as natural populations? The authors should explain where they were originally collected. In Suppl. Table S2, the USDA (or other provider) ID corresponding to each accession should be provided, as it is done in suppl. Table S1.
The authors write that the PCR amplification to investigate presence of the Hopscotch element leads to two amplification products: one for the entire element, (5kb) and one for amplification of part of the element only (1.1kb). Why is this? How do the authors explain the origin of the 1.1kb band? The sentence with “and” on line 83, suggests that the two bands are produced in one single homozygous plant. If some accessions really amplify two bands, is the primer located in the LTR? If it is a typo and the “and” on line 83 should be replaced by a “or”, is the element truncated in some accessions, leading to a 1.1kb band instead of the 5kb one? This should be clarified.
From Figure S1, the middle primer seems to be right in the middle of the element, which suggests it was not designed in the LTR region. Is this true? In which part of the retrotransposon was the primer designed? The authors should add position of the retrotransposon LTRs and coding regions on this figure.
In suppl. Figure S1, the names “HopF/HopR/HopIntR” should be added, as well as corresponding expected amplification sizes.
In suppl. Figure S2, band size of several bands (close in size to these amplified) should be indicated next to the ladder. Legend should explain better what the figure shows. It states “Genotypes are indicated at the top of the gel.” What does this refer to? Numbers? “Hop/Hop” code? The figure should be completely relabeled, so that primer pair names and type of genotype (with presence of absence of Hopscotch) are clearly identified. What is “no Hop/Pif”? Why are there two bands on lane 5? Lane 6 has a weird smear. No primers are visible on lanes 5, 6 and 7.
Bottom gel resolution is poor. On the right for low molecular weight bands, it is difficult to assess that there is a single band (lane 7). A 1% gel and 1kb ladder are clearly not adequate for 300bp band detection.
It is not stated whether PCR products were sequenced to check for correct amplification. Were some of the PCR products sequenced? If not, this should be done.

The authors write: “Environmental data represent average values for the last several decades (climatic data) or are likely stable over time (soil data)”. Does this mean that soil data was not averaged over the last decades? How was it estimated? The fact that soil parameters are “likely” stable over time actually quite depends on what we consider as “soil”. Does this include micro-organisms? A bit more detail should be given.
Figure 1: This figure should show 37 populations of parviglumis and 4 populations of mexicana (see page 8, lines 211 to 213). But the figure legends indicates only parviglumis. It should include mexicana populations, and they should be differentiated by an appropriate color code. In some cases, several circles seem to derive from one circle, suggesting they correspond to several populations from the same location. On the left hand side, it is unexpected to globally have a null frequency of “No Hopscotch” while several populations have a large fraction of “Hopscotch”. This means that, within 25 miles, the frequency of the “Hopscotch” allele varies greatly. The authors should discuss this point. Names of the parviglumis populations used for the rest of the study (La Mesa, San Lorenzo, Ejutla1 and Ejutla2) should be indicated.

Validity of the findings

Results of the PCR amplifications are not well enough described and of high enough quality to be able to conclude whether the retrotransposon detection is valid or not (see “Experimental Design”). The authors should improve labelling of suppl. Figure S2 and, if needed, run the PCR amplicons on a more concentrated gel and with appropriate ladder.
Based on the association genetics study performed, the authors find an absence of association between the Hopscotch insertion and tillering index. At 40 days, they even find a weak but significant correlation, but in the unexpected direction (homozygotes for the Hopscotch insertion have a higher tillering). The authors discuss that this could be due to variation at other unlinked loci. However, they do not discuss on the effect of the environment on tillering. Considering the experiment was performed in greenhouse conditions while teosintes grow in much different environmental conditions, the genotype x environment interaction could differ greatly to this obtained in teosinte natural environmental conditions. This should be discussed.

Additional comments

No additional comments.

---

## Round 0.2 · accepted · Accept

Your findings about the distribution and effects of the hopscotch insertion upstream of tb1 in teosinte raise more questions than they answer. But the conclusions are well supported, and the article is clear and well written. I appreciate your careful attention to the reviewer's comments and the clearly annotated revisions.